# Development and Preliminary Validation of the Sexual Minority Identity Emotion Scale

Jacob Goffnett [1,*], Samantha Robinson [2], Anna Hamaker [3], Mohammod Mahmudur Rahman [2,4,†], Sheree M. Schrager [5] and Jeremy T. Goldbach [6]

1   School of Social Work, Virginia Commonwealth University, Richmond, VA 23284, USA
2   Department of Mathematical Sciences, University of Arkansas, Fayetteville, AR 72701, USA; sewrob@uark.edu (S.R.); mrahman5@kumc.edu (M.M.R.)
3   School of Social Work, University of Arkansas, Fayetteville, AR 72701, USA; achamake@uark.edu
4   Department of Biostatistics & Data Science, University of Kansas Medical Center, Kansas City, KS 66160, USA
5   Department of Graduate Studies and Research, California State University Dominguez Hills, Carson, CA 90747, USA; sschrager@csudh.edu
6   Brown School of Social Work, Washington University in St. Louis, St. Louis, MO 63130, USA; jgoldbach@wustl.edu
*   Correspondence: goffnettj@vcu.edu
†   Mohammod Mahmudur Rahman was affiliated with University of Arkansas at the time of authorship; with University of Kasnas Medical Center at time of publication.

**Abstract:** Emotions influence health behaviors and outcomes, yet little research has examined the emotion–health relationship among sexual minorities. The few studies in this area have used general measures of feelings without regard for identity, despite the literature positing emotions as culturally and contextually specific. This critical limitation obscures inferences made in studies that have found emotions to predict mental health outcomes for sexual minorities. This study begins to address this gap by developing and examining the preliminary validation of the Sexual Minority Identity Emotion Scale, a measure of shame and pride specific to the identity experiences of sexual minority adolescents. The initial pool of items emerged from a qualitative study and was refined through a multistep review. The measurement's factor structure and criterion validity were examined using a nationwide sample of 273 sexual minority adolescents from the United States. The scale has four factors with strong internal reliability, adequate criterion validity, and utility in health research.

**Keywords:** shame; emotions; sexual minority; psychometrics; adolescent; health behavior

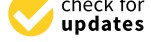



## 1. Introduction and Background

Decades of research demonstrate that emotions influence health outcomes in the general population [1–3], yet they have received little empirical attention in sexual minority health research. Emotions affect health through underlying physiological processes and the behaviors used to manage these feelings [2,4]. Fredrickson's [2] scholarship on positive emotions (e.g., joy, pride, and hope) shows they relate to health through vagal tone, a physiological mechanism of the parasympathetic nervous system that regulates the body during changes. Conversely, negative emotions (e.g., anger, shame, and sadness), are related to health through a variety of physiological changes, such as increased heart rate and hormones related to the sympathetic nervous system [1]. Emotions also elicit behaviors that have implications for health, including increased openness and a social connection (positive emotions) or smoking, alcohol, and other substance use or avoidance and isolation (negative emotions) [1]. Scholars have found the way historically marginalized people express, regulate, and understand their emotions is affected by societal stigma toward their group. For example, adults frequently mistake Black adolescents' emotional expression for anger, consequently leading these adolescents to employ emotional suppression to avoid conflict [5]. Recent theoretical work illustrates the value of examining the emotion–health

relationship among sexual minority people [6–8]. Feinstein [7] posited that anticipatory emotions (e.g., anger, anxiety) may differentially result in internalizing and externalizing behaviors through their interaction with sensitivity to rejection. For example, angry anticipation of rejection may cause externalization resulting in aggression, whereas anxious anticipation of rejection may cause internalization resulting in isolation.

Shame and pride may be particularly salient emotions to sexual minority people in the United States (U.S.) because their identities have been culturally constructed as shameful and something to be hidden or, in some instances, prideful and something to be celebrated [9]. A developing body of empirical research among sexual minority people has detailed how feelings of shame and pride affect their health behaviors and psychosocial well-being [8,10–14]. Quantitative studies using general measures of shame found it to be significantly and positively related to discrimination, concealment of sexuality and internalized stigma, and psychological distress [10,12,15,16]. One study with sexual minority adults found shame to mediate the relationships between discrimination and internalized homonegativity and concealment, respectively, and symptoms of depression and anxiety, in that greater reports of these identity-related stressors were associated with higher feelings of shame, which were in turn associated with greater symptoms of depression and anxiety [12].

Although qualitative research has connected shame to isolation, suicidality, non-suicidal self-injury, and substance misuse among sexual minority youth [8,11], pride has received less research attention regarding health—even in the broader literature. In a recent study of general population adolescents, lower pride and higher shame were associated with poorer self-reported health [17]. Similarly, a study of sexual minority college students found a significant inverse correlation between identity pride and symptoms of anxiety and stress [14]. Qualitatively, pride has been associated with a sustained connection to the queer community [13], a behavior that promotes positive health [18].

The limited body of research on shame, pride, and health among sexual minority people has also been hampered by using general measures of these emotions, failing to account for culturally and contextually specific factors pivotal to emotional construction [1,4,19]. A measure focusing on emotions related to sexual minority identity (i.e., emotions arising because of one's sexual identity) does not exist. The Internalized Homonegativity Inventory for Gay Men [20] measures general feelings among gay men about their identities in relation to societal stigmatization but does not capture discrete emotions, such as shame or pride. One factor included in the scale, the gay affirmation subscale, is composed of seven items that assess the level of positive feelings about and salience of men's sexual minority identities. Woodford et al. [21] modified two items from the gay affirmation subscale—"I am proud to be LGBTQ" and "I believe being LGBTQ is an important part of me"—to create a condensed scale measuring LGBTQ pride. Although the modified scale demonstrates good internal reliability, it does not assess the emotion of pride related to sexual minority identity but rather pride in being a member of the LGBTQ community.

Thus, we assert that a critical barrier to scrupulous emotion–health research among sexual minority people is the lack of standardized measures assessing emotions specific to their identity. Moreover, most emotion–health research to date has relied on adult samples, overlooking earlier developmental periods when life experiences, particularly those related to identity, may lead to long-term emotional construction [22]. Thus, the present study was focused on developing a measure of shame and pride specific to sexual minority adolescents (SMAs) that can be used in future behavioral health research, elucidating potential mechanisms of change for intervention development. As such, we used previously validated measures shown to relate to general feelings of shame or pride among the population, such as internalized homonegativity, self-harm, and substance misuse, to validate the measure's concurrent validity.

## 2. Materials and Methods

### 2.1. Step 1: Item Development

Item development was guided by the construct definitions of shame and pride resulting from a prior qualitative study conducted by the first author [8]. The study used grounded theory methods to analyze life history interviews with 36 SMAs to understand the psychological, social, and behavioral manifestations of shame and pride and their antecedents and consequences. The construct definitions were refined further during the expert review discussed in the next section. Shame was defined as a belief that one's sexual identity is worthless, accompanied by overwhelming feelings of displeasure and a motivation to hide the identity. Pride was defined as a belief that one's sexual identity has value, accompanied by feelings of zestful pleasure and a motivation to continue developing the identity through connections. Thus, the study found shame and pride to have three dimensions each that reflected beliefs, feelings, and goals. An initial pool of 43 items (21 for shame, 22 for pride) was developed to reflect the 3 highlighted dimensions (beliefs, feelings, and goals) that emerged for both shame and pride.

We used an experience-based approach to assessing feelings of shame and pride related to sexual minority identity, considering extant measurement research and findings from a preliminary qualitative study. We curated a list of shame and pride experiences related to sexual minority identity that are general enough to not depreciate and can be easily administered in various settings. Item construction followed measurement standards, including the use of clear language relevant to adolescents, avoiding presumptive or leading verbiage, and constructing items that would require different levels of shame or pride [23,24]. Table 1 displays the initial pool of items.

**Table 1.** Initial pool of items for the Sexual Minority Identity Emotion Scale.

| | |
|---|---|
| 1. | My sexual orientation makes me feel abnormal. |
| 2. | I am a deeply flawed person because of my sexual orientation. |
| 3. | When my sexual orientation is at risk of being exposed I want to hide. |
| 4. | I feel like an alien because of my sexual orientation. |
| 5. | I believe my sexual orientation makes me not as valuable as other people. |
| 6. | I want to make myself as small as possible when someone mentions LGBTQ+ people. |
| 7. | I do not feel fully human at times because of my sexual orientation. |
| 8. | I believe I am not as worthy as other people because of my sexual orientation. |
| 9. | I wish I could make myself invisible because of my sexual orientation. |
| 10. | I try to reduce my presence in social situations where my sexual orientation is at risk of being exposed. |
| 11. | I try to avoid media featuring LGBTQ+ content while with someone that does not know my sexual orientation. |
| 12. | In general, I feel like a disappointment because of my sexual orientation. |
| 13. | I am not as good of a person as my heterosexual peers because of my sexual orientation. |
| 14. | I start to retreat when my sexual orientation starts to feel too visible. |
| 15. | In general, I feel unimportant because of my sexual orientation. |
| 16. | I shy away from religious situations because they make me feel like a flawed human. |
| 17. | I believe my sexual orientation makes me a damaged person. |
| 18. | I try to get out of a situation when my sexual orientation is starting to show. |
| 19. | My sexual orientation makes me feel wrong. |
| 20. | My sexual orientation makes me a broken person. |
| 21. | I get overwhelmed with feelings when others can tell that I am not straight. |
| 22. | Based on messages I receive from others, my sexual orientation feels valuable. |
| 23. | I think my sexual orientation makes me a more authentic person. |
| 24. | My sexual orientation makes me more of who I am. |
| 25. | I am motivated to be more open with my sexual orientation with others. |
| 26. | Sharing my sexual orientation with others makes me feel successful. |
| 27. | I feel like my sexual orientation is important. |
| 28. | My sexual orientation is a core piece of who I am. |
| 29. | I want to be more connected to the LGBTQ+ community. |
| 30. | Connecting to the LGBTQ+ community makes my sexual orientation feel valuable. |

**Table 1.** *Cont.*

| | |
|---|---|
| 31. | I feel satisfied because of my sexual orientation. |
| 32. | I am certain about who I am thanks to my sexual orientation. |
| 33. | I am eager to learn new things related to my sexual orientation. |
| 34. | I embrace learning new things about my sexual orientation. |
| 35. | My sexual orientation feels right to me. |
| 36. | My sexual orientation helps me feel more complete. |
| 37. | I feel self-assured when others give positive acknowledgment to my sexual orientation. |
| 38. | I want others in the LGBTQ+ community to feel valid. |
| 39. | I feel like my sexual orientation is valid. |
| 40. | My sexual orientation gives me confidence. |
| 41. | I am inspired to connect with people who affirm my sexual orientation. |
| 42. | I want to support my peers who are also a part of the LGBTQ+ community. |
| 43. | I totally accept my sexual orientation. |

*2.2. Step 2: Measurement Review*

The items underwent a three-step review to refine the measure before being administered for testing. In the first review, two graduate students, one in family sciences and one in public health, read through each item with the researcher to evaluate its adherence to measurement standards. During this step, items were modified to improve language and structure. Because both reviewers had substantive knowledge of sexual minority people, they also provided feedback on the item content. This process helped create sound items, reducing the cognitive burden in subsequent reviews. Next, five experts in either SMA well-being or social emotions (i.e., shame and pride) provided feedback on the items, construct definitions, and instructional prompts provided to respondents. All components were modified based on expert feedback. One item was dropped because most experts rated it as poor and provided consistent feedback that it did not fit alongside the other items (i.e., "I shy away from religious situations because they make me feel like a flawed human").

In the final review step, five SMAs participated in cognitive interviews, where they went through each item with the researcher, provided a response to the item, and discussed their thought process in responding [23]. Participants were between 16 and 19 years old and were recruited from a local community agency serving sexual minority people. Parental consent was waived to protect participants from having to disclose their identities to their parents to participate in the study [25]. A university institutional review board approved procedures for the cognitive interviews but waived oversight for the first two steps of the review.

*2.3. Step 3: Preliminary Modeling and Validity*

2.3.1. Participants and Recruitment

Data were collected for the refined 42-item Sexual Minority Identity Emotion Scale (SMIES) as part of a longitudinal study examining minority stress over time in a national sample of SMAs. Eligibility criteria at baseline included being between 14 and 17 years old, identifying as a cisgender male or female, residing in the United States, and identifying as not 100% heterosexual. Targeted advertisements were disseminated on popular social media platforms to gather a demographically diverse sample. Sampling was stratified by five geographic regions of the United States (i.e., Northeast, Southeast, Midwest, Southwest, West), sex, and urbanization (i.e., urban or rural), resulting in 20 targeted groups. Respondent-driven sampling—a form of snowball sampling—was also used with eligible participants, asking them to identify individuals in their social network who might be interested in completing the study. Participants received a USD 10 incentive for each friend who completed the survey.

The social media advertisements contained a link that directed potential participants to Qualtrics, where they completed the eligibility screener, informed consent, and baseline survey. After completing the survey, participants were asked if they were interested in

participating in a longitudinal survey and, if so, to provide methods for future contact. Participants who completed the entire survey received USD 15 for their participation. Participants who showed interest in the follow-up surveys were contacted in 6-month increments from baseline. Participants were removed from the study if they provided low-quality data at any of the 6-month waves (e.g., unrealistically short completion time, low correct response to validation questions, and moderately high rates of "decline to answer"). Data for the current study were collected at the 18-month follow-up (N = 273). After completing the main study at this follow-up, participants were asked if they wanted to complete an additional short survey containing the shame and pride items and general measures of shame and pride for testing concurrent criterion validity. Only a subset of participants from the main study completed the additional short survey. A university institutional review board approved the study procedures.

Participants' mean age was 17.4 years (SD = 1.1 years). Most participants identified their gender as female (63%) or male (27%). Bisexual (41%), gay (16%), and lesbian (16%) were the most common sexual orientations. More than half of the sample was White (56%), followed by multiracial (14%) and Latinx or Hispanic (13%) participants. Finally, 34% of the sample had completed the 11th grade, and 40% were high school graduates, had completed a GED, or had completed some college. See Table 2 for full participant characteristics.

**Table 2.** Sociodemographic characteristics of survey participants (*n* = 273).

| Characteristic | *n* | % |
|---|---|---|
| Age (years) | | |
| 15 | 9 | 3.3 |
| 16 | 43 | 15.8 |
| 17 | 88 | 32.2 |
| 18 | 87 | 31.9 |
| 19 | 45 | 16.5 |
| Race | | |
| Asian or Pacific Islander | 20 | 7.3 |
| Black or African American | 20 | 7.3 |
| Latinx or Hispanic | 34 | 12.5 |
| Multiracial | 37 | 13.6 |
| Native American, American Indian, or Alaska Native | 10 | 3.7 |
| White | 152 | 55.7 |
| Sex assigned at birth | | |
| Female | 197 | 72.2 |
| Male | 76 | 27.8 |
| Gender identity | | |
| Female | 172 | 63.0 |
| Male | 74 | 27.1 |
| Genderqueer or gender nonconforming | 5 | 1.8 |
| Nonbinary | 14 | 5.1 |
| Trans male | 6 | 2.2 |
| Not answered | 2 | 0.7 |
| Sexual orientation | | |
| Asexual | 9 | 3.3 |
| Bisexual | 113 | 41.4 |
| Demisexual | 2 | 0.7 |
| Gay | 44 | 16.1 |
| Homosexual | 5 | 1.8 |
| Lesbian | 43 | 15.8 |
| Pansexual | 25 | 9.2 |
| Queer | 18 | 6.6 |
| Questioning | 9 | 3.3 |
| Other | 5 | 1.8 |

**Table 2.** *Cont.*

| Characteristic | *n* | % |
|---|---|---|
| Education completed | | |
| 9th grade or less | 16 | 5.9 |
| 10th grade | 51 | 18.7 |
| 11th grade | 94 | 34.4 |
| High school graduate or GED | 50 | 18.3 |
| Trade school or some college | 62 | 22.7 |
| Employment | | |
| No, never | 147 | 53.8 |
| No, but had previous job | 107 | 39.2 |
| Yes, part time | 18 | 6.6 |
| Not answered | 1 | 0.4 |
| Region | | |
| West | 58 | 21.2 |
| Southwest | 44 | 16.1 |
| Midwest | 36 | 13.2 |
| Southeast | 64 | 23.4 |
| Northeast | 71 | 26.0 |

### 2.3.2. Measures

Participant demographic characteristics were collected as part of the main longitudinal survey at baseline, and some characteristics were also asked at each follow-up to capture changes (i.e., age, gender, sexual orientation, religion and spirituality, education, ZIP code). Additionally, participants completed standardized measures of internalized homonegativity, expectations of rejection, anxiety, and depression, and single-item assessments of self-harm and prescription pain reliever use without a doctor's orders, as part of the main survey. These self-report measures were used to assess the SMIES' criterion validity. We hypothesized that identity-related shame would have a positive correlation with proximal minority stressors, behavioral health indicators, and general feelings of shame and an inverse correlation with general feelings of pride. Conversely, we hypothesized that identity-related pride would have an inverse correlation with proximal minority stressors, behavioral health indicators, and general feelings of shame and a positive correlation with general feelings of pride.

### Proximal Minority Stressors

Two subscales of the Sexual Minority Adolescent Stress Inventory (SMASI) [26,27] were used to assess internalized homonegativity and negative expectancies. Participants were asked since the last time they completed the survey if they had experienced seven items assessing internalized homonegativity and three items assessing expectations of rejection (answered as a dichotomous "yes" or "no"). Subscales were scored as percentages of items endorsed (e.g., 33%, 66%, or 100% for the three items of the expectations of rejection subscale).

### Behavioral Health

The 7-item Generalized Anxiety Disorder scale [28] was used to assess anxiety symptoms during the past 2 weeks. Participants indicated how frequently they experienced a symptom on a 4-point Likert scale (0 = not at all, 1 = several days, 2 = more than half the days, 3 = nearly every day). Symptoms of depression were assessed using the Center for Epidemiologic Studies Depression scale [29]. The 4-item version asks participants how frequently in the past week they experienced a symptom of depression on a 4-point Likert scale (0 = rarely or none of the time [less than one day], 1 = some or a little of the time [1–2 days], 2 = occasionally or a moderate amount of time [3–4 days], 3 = most or all of the time [5–7 days]). Self-harm (e.g., "During the past 6 months, how many times did you do something to purposely hurt yourself without wanting to die, such as cutting or burning

yourself on purpose?") and prescription pain reliever use without a doctor's orders (i.e., "Since the last time you took this survey, have you used prescription pain relievers (e.g., Vicodin and OxyContin) without a doctor's orders?") were assessed with single items from the Centers for Disease Control and Prevention's [30] Youth Risk Behavior Survey. Participants provided a yes-or-no response to each item.

SMIES

Participants were prompted to consider their feelings and beliefs during the past month while rating their level of agreement with a sexual identity-related shame or pride item on a 5-point Likert scale (1 = strongly disagree, 2 = disagree, 3 = neither agree nor disagree, 4 = agree, 5 = strongly agree).

General Shame and Pride

General shame was assessed with items from the Test of Self-Conscious Affect– Adolescent (TOSCA-A) [31]. The TOSCA-A is a scenario-based measure that provides situations that may elicit an emotion among adolescents and asks participants to indicate how likely they are to have a shame response on a 5-point Likert scale (1 = not at all likely, 2 = unlikely, 3 = maybe, half and half, 4 = likely, 5 = very likely). The Authentic Pride Scale (APS) [32] was used to assess general feelings of pride. The APS asks participants to indicate, on average, the extent to which they experience 7 adjectives related to pride (e.g., "successful") on a 5-point Likert scale (1 = not at all, 2 = somewhat, 3 = moderately, 4 = very much, 5 = extremely).

### 2.3.3. Data Analysis

All analyses were conducted using R version 4.1.1 [33]. Prior to analysis, the study data from 273 SMAs were screened for univariate outliers, and the factorability of the 42 SMIES items was assessed. The Kaiser–Meyer–Olkin measure of sampling adequacy was 0.93, exceeding the suggested cutoff of 0.60 [34], and Bartlett's test of sphericity was significant, suggesting that factor analysis was suitable for the SMIES items. Exploratory factor analysis, internal consistency reliability, and criterion validity analyses were conducted. Additionally, correlational analyses were used to examine the relationship between behavioral health outcomes and subscales of the SMIES with statistical significance defined as $p < 0.05$.

## 3. Results
### 3.1. Exploratory Factor Analysis

Principal axis factor extraction with a direct oblimin (oblique) rotation of the SMIES items was conducted on the item responses from the 273 participants. An oblique rotation method was selected for the analysis to improve factor interpretability and achieve a simple structure because the factors should theoretically relate to the latent constructs of shame and pride and intercorrelate [23,35]. To determine the number of factors to retain, Kaiser's [34] criterion, which suggests retaining factors with an eigenvalue greater than 1, was used in conjunction with Cattell's [36] scree test, the optimal coordinate method, Velicer's [37] minimum average partial test, and parallel analysis [38], as implemented in the nFactors package in R [39]. Criteria for item retention were also set a priori based on Tabachnick and Fidell's [35] guidelines. Items with a moderate loading (≥0.50) onto one factor were retained, whereas all items that did not moderately load onto one factor were eliminated.

Based on these analyses, the optimal number of factors to retain was four. After fitting a 4-factor solution of the 42 items and examining item factor loadings, 7 items were eliminated due to poor loadings (<0.50). The remaining 35 items contributed to a simple factor structure and had moderate loadings on one primary factor (see Table 3). Items were grouped into subscales based on their primary loadings on the four factors: shame concepts, shame goals, pride concepts, and pride goals. Shame and pride concepts include items assessing beliefs and feelings about one's sexual identity. Shame and pride goals include items that assess behaviors stemming from these emotions. The overall shame measure had

19 items with a potential score range of 19–95. The shame concepts subscale has 12 items with a potential score range of 12–60 and the shame goals subscale has 7 items with a potential score range of 7–35. The overall pride measure had 16 items with a potential score range of 16–80. The pride concepts subscale has 11 items with a potential score range of 11–55 and the pride goals subscale has 5 items with a potential score range of 5–25.

**Table 3.** Item means, standard deviations, and factor loadings ($\hat{\lambda}$) with subscale internal consistency reliability.

| SMIES Item | M | SD | F1 $\hat{\lambda}$ | F2 $\hat{\lambda}$ | F3 $\hat{\lambda}$ | F4 $\hat{\lambda}$ |
|---|---|---|---|---|---|---|
| **Factor 1: Shame concepts ($\alpha$ = 0.93; $\omega$ = 0.94)** | | | | | | |
| 1. My sexual orientation makes me feel abnormal. | 2.32 | 1.24 | **0.52** | −0.11 | 0.17 | 0.12 |
| 2. I believe I am a deeply flawed person because of my sexual orientation. | 1.66 | 0.95 | **0.86** | 0.01 | −0.06 | −0.06 |
| 3. I feel defective because of my sexual orientation. | 1.84 | 1.06 | **0.74** | −0.11 | 0.05 | 0.08 |
| 4. I think my sexual orientation makes me less valuable than other people. | 1.72 | 0.91 | **0.63** | −0.01 | 0.02 | −0.06 |
| 5. I do not feel fully human at times because of my sexual orientation. | 1.70 | 0.96 | **0.64** | 0.13 | 0.12 | 0.00 |
| 6. I believe I am not as worthy as other people because of my sexual orientation. | 1.72 | 0.96 | **0.77** | 0.06 | 0.01 | −0.04 |
| 7. I feel like a disappointment because of my sexual orientation. | 2.14 | 1.21 | **0.50** | −0.04 | 0.16 | 0.05 |
| 8. I think I am not as good of a person as my straight peers because of my sexual orientation. | 1.62 | 0.90 | **0.79** | 0.09 | −0.07 | −0.13 |
| 9. I feel inferior to others because of my sexual orientation. | 1.86 | 1.08 | **0.69** | 0.02 | 0.13 | −0.04 |
| 10. I believe my sexual orientation makes me a damaged person. | 1.73 | 0.94 | **0.79** | −0.01 | −0.02 | −0.01 |
| 11. My sexual orientation makes me feel wrong. | 1.87 | 1.05 | **0.63** | −0.17 | 0.11 | 0.02 |
| 12. I think my sexual orientation makes me a broken person. | 1.70 | 0.93 | **0.79** | −0.10 | 0.00 | 0.04 |
| **Factor 2: Pride concepts ($\alpha$ = 0.90; $\omega$ = 0.92)** | | | | | | |
| 1. I feel my sexual orientation adds value to my life. | 3.64 | 1.00 | −0.08 | **0.62** | −0.01 | 0.19 |
| 2. I think my sexual orientation makes me a more authentic person. | 3.75 | 0.92 | 0.08 | **0.52** | 0.05 | 0.16 |
| 3. My sexual orientation makes me more of who I am. | 3.94 | 0.96 | 0.08 | **0.62** | −0.05 | 0.15 |
| 4. My sexual orientation is a core piece of who I am. | 3.68 | 1.10 | 0.02 | **0.61** | 0.06 | 0.15 |
| 5. My sexual orientation gives me a sense of purpose. | 3.18 | 1.08 | 0.17 | **0.64** | −0.05 | 0.13 |
| 6. I am certain about who I am because of my sexual orientation. | 3.39 | 1.12 | 0.03 | **0.79** | 0.01 | −0.10 |
| 7. My sexual orientation feels right to me. | 4.12 | 0.87 | −0.23 | **0.61** | −0.10 | −0.11 |
| 8. My sexual orientation helps me feel more complete. | 3.62 | 1.01 | −0.01 | **0.69** | 0.02 | 0.11 |
| 9. I feel like my sexual orientation is correct. | 4.05 | 0.90 | −0.22 | **0.62** | 0.02 | −0.19 |
| 10. My sexual orientation gives me confidence. | 3.33 | 1.02 | −0.04 | **0.58** | −0.14 | 0.09 |
| 11. I totally accept my sexual orientation. | 3.94 | 1.14 | −0.23 | **0.50** | −0.18 | −0.12 |
| **Factor 3: Shame goals ($\alpha$ = 0.90; $\omega$ = 0.93)** | | | | | | |
| 1. When my sexual orientation is at risk of being exposed I want to hide. | 2.80 | 1.30 | −0.04 | 0.00 | **0.81** | −0.02 |
| 2. I want to make myself as small as possible when someone mentions LGBTQ+ people. | 2.15 | 1.15 | 0.11 | 0.06 | **0.63** | −0.12 |
| 3. I try to tone myself down in social situations where my sexual orientation is at risk of being exposed. | 2.90 | 1.30 | −0.05 | 0.04 | **0.84** | 0.03 |
| 4. I try to avoid media featuring LGBTQ+ content when I'm with someone that does not know my sexual orientation. | 2.53 | 1.37 | −0.10 | 0.06 | **0.68** | 0.05 |
| 5. I start to retreat when my sexual orientation starts to feel too visible. | 2.60 | 1.24 | 0.10 | −0.05 | **0.77** | 0.01 |
| 6. I try to get out of a situation when my sexual orientation is starting to show. | 2.41 | 1.21 | 0.02 | −0.04 | **0.80** | −0.01 |
| 7. I get overwhelmed with uncomfortable feelings when others can tell I am not straight. | 2.59 | 1.25 | 0.08 | −0.05 | **0.71** | −0.03 |
| **Factor 4: Pride goals ($\alpha$ = 0.78; $\omega$ = 0.84)** | | | | | | |
| 1. I want to be more connected to the LGBTQ+ community. | 4.07 | 0.93 | 0.01 | 0.24 | −0.01 | **0.52** |
| 2. Connecting to the LGBTQ+ community makes my sexual orientation feel valuable. | 3.94 | 0.99 | 0.01 | 0.29 | 0.03 | **0.50** |
| 3. I am eager to learn new things related to my sexual orientation. | 4.05 | 0.90 | −0.06 | 0.05 | −0.01 | **0.73** |
| 4. I embrace learning new things about my sexual orientation. | 4.16 | 0.78 | −0.13 | 0.00 | −0.04 | **0.66** |
| 5. I want to support my peers who are also a part of the LGBTQ+ community. | 4.69 | 0.58 | −0.04 | 0.07 | −0.01 | **0.51** |

**Note**: Significant factor loadings are in boldface.

The factor correlation matrix was examined to determine if the oblique rotation was appropriate for the data in that factors correlated at 0.32 or higher [35]; three of the six-

factor correlations met this criterion, with two factors correlating at 0.60 or higher. Oblique rotation was retained in light of these findings.

### 3.2. Internal Consistency Reliability Analysis

Internal consistency reliability for each subscale (shame concepts, shame goals, pride concepts, and pride goals) was examined using both Cronbach's coefficient alpha and coefficient omega total, a less biased estimate of reliability recommended by McNeish [40] and Trizano-Hermosilla et al. [41]. Reliabilities for each subscale are reported in Table 3. Descriptive statistics for each subscale and the complete set of shame and pride items are reported in Table 4. Higher scores indicated greater levels of the respective latent trait.

**Table 4.** Descriptive statistics and criterion validity.

|  | **M** | **SD** | **TOSCA-A** [a] | **APS** [b] | **SMASI-IH** [c] | **SMASI-NE** [d] |
|---|---|---|---|---|---|---|
| Shame total | 2.10 | 0.76 | 0.40 *** | −0.32 *** | 0.56 *** | 0.53 *** |
| Shame concepts | 1.82 | 0.77 | 0.34 *** | −0.26 *** | 0.59 *** | 0.44 *** |
| Shame goals | 2.57 | 1.00 | 0.38 *** | −0.32 *** | 0.38 *** | 0.52 *** |
| Pride total | 3.85 | 0.61 | −0.15 * | 0.30 *** | −0.44 *** | −0.07 |
| Pride concepts | 3.69 | 0.71 | −0.17 ** | 0.33 *** | −0.47 *** | −0.13 * |
| Pride goals | 4.18 | 0.61 | −0.03 | 0.10 | −0.20 ** | 0.11 |

[a] Test of Self-Conscious Affect–Adolescent Shame, subscale; [b] Authentic Pride Scale; [c] Sexual Minority Adolescent Stress Inventory, internalized homonegativity subscale; [d] Sexual Minority Adolescent Stress Inventory, negative expectancies subscale; * $p < 0.05$. ** $p < 0.01$. *** $p < 0.001$.

Cronbach's alpha values were all moderate to high. The shame items displayed high internal consistency for both the shame concepts ($\alpha = 0.93$, 95% CI: [0.92, 0.94]; $\omega = 0.94$, 95% CI: [0.91, 0.94]) and shame goals ($\alpha = 0.90$, 95% CI: [0.88, 0.92]; $\omega = 0.93$, 95% CI: [0.91, 0.95]) subscales. The pride items also displayed moderate to high internal consistency for the subscales of pride concepts ($\alpha = 0.90$, 95% CI: [0.88, 0.91]; $\omega = 0.92$, 95% CI: [0.91, 0.93]) and pride goals ($\alpha = 0.78$, 95% CI: [0.73, 0.81]; $\omega = 0.84$, 95% CI: [0.79, 0.89]). The scales and subscales of the SMIES exhibited adequate internal consistency. Scale calculations with each item deleted were also analyzed and suggested that removing items would statistically weaken the scales; thus, all remaining items were retained. No substantial increases in internal consistency reliability, as measured by Cronbach's alpha, could have been achieved by removing more items.

### 3.3. Criterion Validity Analysis

Concurrent criterion validity was assessed by analyzing the bivariate relationships between the SMIES scales and subscales and all validation measures. Specifically, the correlations of SMIES scores with scores on the TOSCA-A, APS, and SMASI subscales were assessed. A priori guidelines were used to interpret convergent validity, with 0.00 to 0.30 indicating negligible correlations, 0.30 to 0.50 indicating low correlations, 0.50 to 0.70 indicating moderate correlations, and 0.70 and above to be high correlations [42]. Correlation coefficients between SMIES subscale scores and scores on the TOSCA-A, APS, and SMASI subscales are displayed in Table 4, along with corresponding two-tailed $p$-values.

The SMIES shame scales and the TOSCA-A had low and statistically significant correlations: shame total ($r = 0.40$, $p < 0.001$), shame concepts ($r = 0.34$, $p < 0.001$), and shame goals ($r = 0.38$, $p < 0.001$). Correlations between the SMIES shame scales and internalized homonegativity and negative expectancies subscales of the SMASI were also statistically significant at low to moderate levels: for internalized homonegativity, shame total ($r = 0.55$, $p < 0.001$), shame concepts ($r = 0.58$, $p < 0.001$), and shame goals ($r = 0.37$, $p < 0.001$); for negative expectancies, shame total ($r = 0.53$, $p < 0.001$), shame concepts ($r = 0.44$, $p < 0.001$), and shame goals ($r = 0.52$, $p < 0.001$). The SMIES shame scales had a significant inverse relation at negligible to low levels to the APS: shame total ($r = −0.32$, $p < 0.001$), shame concepts ($r = −0.26$, $p < 0.001$), and shame goals ($r = −0.32$, $p < 0.001$).

The SMIES pride total scale and pride concepts subscale had low and statistically significant correlations with the APS measure of pride: pride total ($r = 0.30$, $p < 0.001$) and pride concepts ($r = 0.33$, $p < 0.001$). Negative correlations between the SMIES pride scales and internalized homonegativity of the SMASI were all statistically significant at negligible to low levels: pride total ($r = -0.44$, $p < 0.001$), pride concepts ($r = -0.46$, $p < 0.001$), and pride goals ($r = -0.22$, $p < 0.001$). The pride concepts subscale of the SMIES had a significant negative correlation at a negligible level with the negative expectancies subscale of the SMASI ($r = -0.13$, $p < 0.05$). Correlations between the SMIES pride scales and the TOSCA-A were negligible, with pride total ($r = -0.15$, $p = 0.014$) and pride concepts ($r = -0.17$, $p = 0.004$) having statistically significant negative relationships.

### 3.4. Correlational Analysis with Behavioral Health Outcomes

Correlational analyses were used to examine the relationships between behavioral health outcomes and the SMIES shame and pride items. Composite scores for each participant for overall shame and pride scales and each subscale (shame concepts, shame goals, pride concepts, and pride goals) were calculated by averaging the responses for those scale items. The correlation between these composite scores and various behavioral health outcomes is displayed in Table 5, along with the associated two-tailed *p*-values.

**Table 5.** Concurrent validity and correlation with behavioral health outcomes.

| | GAD-7 [a] | CES-D [b] | Self-Harm | Pain Rx |
|---|---|---|---|---|
| Shame total | 0.24 *** | 0.27 *** | 0.20 ** | 0.17 ** |
| Shame concepts | 0.22 *** | 0.23 *** | 0.19 ** | 0.12 |
| Shame goals | 0.20 *** | 0.23 *** | 0.17 ** | 0.21 *** |
| Pride total | −0.11 | −0.05 | −0.14 * | 0.07 |
| Pride concepts | −0.13 * | −0.07 | −0.14 * | 0.05 |
| Pride goals | −0.01 | 0.03 | −0.09 | 0.10 |

[a] Generalized Anxiety Disorder 7 scale; [b] Center for Epidemiologic Studies Depression scale; * $p < 0.05$. ** $p < 0.01$. *** $p < 0.001$.

Results indicated that shame composite scores had negligible to low associations with anxiety, depression, reports of self-harm, and nonprescribed pain reliever use, with statistically significant correlations ranging from 0.17 to 0.27 (see Table 5). The results demonstrated that higher levels of shame correlated with significantly worse reported health outcomes. Results also indicated that pride concepts composite scores had inverse and significant associations with anxiety ($r = -0.13$, $p = 0.036$) and reports of self-harm ($r = -0.14$, $p = 0.020$) at a negligible level.

## 4. Discussion

The aim of this study was to develop a measure of shame and pride specific to SMAs to be used in health research. Because no previous measures of emotion relating to sexual identity existed, we utilized a multifaced approach to develop the SMIES, simultaneously attending to the limitations of existing measures. A recent systematic review of general shame measures found a lack of detailed developmental studies, inconsistent nominal definitions, and low content validity [19]. A major barrier to defining shame and pride is the general approach to measurement because these emotions are deeply personal and highly contextualized [4,19,43]. We addressed this limitation by specifying a target population and health-based context. Relatedly, this begins to address the content validity limitations of past measures [19]. To further address content validity, we rooted our construct definitions and item pool in a preliminary grounded theory study, the extant literature, and conventions of measurement development, and utilized a multistep review process [8,23].

Findings from the qualitative study from which the SMIES emerged demonstrated the conditions relevant to feelings of shame and pride among sexual minority adolescents. The importance of beliefs and goals to the construction of these emotions was highlighted in the three dimension definition that guided the development of the measure's items [8]. As

such, the measure aimed to assess the construction of shame and pride through feelings and related beliefs and goals. Many theories of emotions highlight the importance of beliefs (cognitions) or goals (behaviors) to emotional generation and regulation [4,44]. By capturing the construction of shame and pride our measure brings beliefs and goals more central to the assessment of shame and pride deviating slightly from past approaches that consider these discrete processes. Exploratory factor analysis revealed a four-factor structure with two dimensions each for shame and pride that reflected concepts and goals. The shame and pride concepts dimension combined items intended to assess feelings and beliefs separately. In American English, "beliefs" and "feelings" may be used synonymously, engendering similar response patterns to these items. Furthermore, a contemporary theory of emotions proposes that beliefs and feelings are components of the neurophysiological cascade that leads to emotional construction [4], which may account for the significant overlap of these dimensions. The four-factor structure of the SMIES has theoretical and practical support and displays strong internal consistency among its subscales.

The measure also displayed satisfactory criterion validity between subscales of the SMIES and general measures of shame and pride. SMIES shame scales and TOSCA-A, a general measure of shame, had low correlations ($r = 0.34$–$0.40$). Rizvi's [43] development of the Shame Inventory, a general measure of shame, assessed convergent validity against other general shame measures, including an adult version of the TOSCA, finding low correlations ($r = 0.37$–$0.50$). It is not surprising the SMIES shame scales would have slightly weaker correlations because they reflect an identity-specific assessment of shame. Similarly, we found low correlations between the total pride scale and the pride concepts subscale of the SMIES and the APS [32]. APS is an adjective checklist that lacks context and goals that are posited to be important to the construction of emotion [4]; the diverging composition between the SMIES and the APS may account for the low and nonsignificant correlations between the APS and pride goals.

Findings from this study demonstrate significant relationships between proximal minority stressors of internalized homonegativity and negative expectancies and identity-related shame and pride. The significant correlations between the internalized homonegativity measure and the SMIES are consistent with the literature on shame and pride in that these emotions are personal and contingent on identity, particularly how one perceives their identity in relation to societal norms [43,45,46]. Allen and Oleson [15] proposed that internalized homonegativity is the "introjection of society's negative attitudes" toward sexual minorities, whereas shame is the "intrapsychic experience of failing to meet an internalized ideal" (p. 34). These researchers examined the relationship between shame and internalized homonegativity, finding a significant, albeit low, correlation ($r = 0.30$). Findings from this study provide further support for shame and internalized homonegativity as distinct but related constructs. These findings also provide further impetus for examining the interactions between emotions and internalized homonegativity and negative expectancies, as proposed by Feinstein [7]. Shame had moderate correlations with negative expectancies. Scholars have described shame as a primary social emotion that is deeply painful because the anticipation of shame engenders social isolation [45,47,48]. Perhaps examining interactions of emotional components and internalized homonegativity and negative expectancies can enhance our understanding of health behaviors and outcomes among sexual minority people.

Our results demonstrate preliminary support for the utility of SMIES in health research. Using an identity-specific measure of shame, we identified similar relationships between shame, anxiety, and depression in past quantitative studies of sexual minority adults [12,16]. We also provided correlational support for qualitative studies that have connected shame among SMAs to self-harm and substance misuse [8,11]. As for pride, our results begin to establish an area of research that has received little attention. We found a significant inverse relationship between pride concepts and anxiety symptoms, like Woodford et al.'s [14] study using a measure of pride regarding membership in the LGBTQ community. Nonsignificant or low correlations require further investigation. Emotions are temporal feelings that give

direction to thoughts and behaviors [44]—often, these behaviors have health implications. Testing the indirect effects of emotion on health outcomes, such as through health behaviors, may better reflect the role of shame and pride in health outcomes for sexual minority folks. The relationship between identity-related shame and pride and behavioral health outcomes provides support for intervention work that reduces the impact of shame and increases the influence of pride.

## 5. Limitations

Although this study contributes a novel measure of shame and pride specific to the experiences of SMAs and opens new avenues of research, some limitations should be noted. Our sampling methods are useful for recruiting marginalized people who are geographically dispersed [49] but influence the generalizability and interpretation of findings. We sampled only those with access to the internet and who engage with social media. Furthermore, respondent-driven sampling can create homogeneity from the sample including like-minded individuals. This study used a classical measurement model that evaluates the overall quality of the measure but not the quality of each item, as is the case with item response theory models [24]. Future studies should conduct discrimination and difficulty tests on the measure's items to provide a better understanding of its construction. The full reliability and validity of the scale cannot be inferred from this study alone because only one aspect was tested for each domain. Additional reliability and validity tests are needed to examine the construction and utility of the SMIES. For example, Lear et al.'s [19] review of shame measures noted a paucity of invariance testing to discern which group's measures perform better.

## 6. Conclusions

Shame and pride about one's sexual minority identity are valid constructs discrete from general shame and pride and internalized homonegativity, as demonstrated by the SMIES. The SMIES needs additional testing to continually improve and understand its construction, reliability, and validity. Furthermore, future research should use the SMIES to understand these emotions in the context of the minority stress framework. Doing so may provide new insights into minority stress processes and health behaviors and outcomes among SMAs. Practitioners should consider helping clients reduce shame and increase pride because our findings correlate these emotions with behavioral health.

**Author Contributions:** Conceptualization, J.G.; Methodology, J.G., S.M.S. and J.T.G.; Formal Analysis, J.G., S.R. and M.M.R.; Resources, J.T.G. and S.M.S.; Data Curation, S.M.S. and J.T.G.; Writing—Original Draft Preparation, J.G. and A.H.; Writing—Review and Editing, S.R., S.M.S. and J.T.G.; Funding Acquisition, S.M.S. and J.T.G. All authors have read and agreed to the published version of the manuscript.

**Funding:** The funding for survey data collection was provided by the National Institute of Minority Health and Health Disparities (NIMHD-5R01MD012252-03; PI: Goldbach).

**Institutional Review Board Statement:** The study was conducted in accordance with the Declaration of Helsinki, and approved by the Institutional Review Boards of the University of Illinois at Urbana-Champaign (#202212035) and Washington University at St. Louis (#202212035).

**Informed Consent Statement:** Informed consent was obtained from all subjects involved in the study.

**Data Availability Statement:** Data reports and datasets can be obtained by emailing the co-author and coordinating a data sharing agreement, Jeremy T. Goldbach (jgoldbach@wustl.edu).

**Acknowledgments:** The first author thanks his dissertation committee for their assistance with this study: Liliane Windsor, Kate Wegmann, Janet Liechty, and Megan Paceley.

**Conflicts of Interest:** The authors declare no potential conflicts of interest with respect to the research, authorship, and/or publication of this article.

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
