# Peer review of "Development and Preliminary Validation of the Sexual Minority Identity Emotion Scale"

_adolescents, doi:10.3390/adolescents4010012_

Round 1
Reviewer 1 Report
Comments and Suggestions for Authors
The paper entitled: “Development and Preliminary Validation of the Sexual Minority Identity Emotion Scale” aims to develop a measure of shame and pride specific to sexual minority adolescents to be used in health research. At the methodological level, this study was conducted in three steps. In the first step (Item development), the authors constructed the definitions of pride and shame based on a prior qualitative study conducted by one of the authors. In the second step (Measurement review) the authors underwent a three-step review of an initial pool of 43 items (21 for shame, 22 for pride) to refine the measure before being administered for testing. Finally, in the third step (Preliminary Modeling and Validity) data were collected during 18 months for the refined 42-item Sexual Minority Identity Emotion Scale. 273 sexual minority adolescents participated in this study. To have a demographically diverse sample the authors stratified the sample in five geographic regions of the United States, according to gender and urbanization.
The authors concluded that shame composite scores had negligible to low associations with anxiety, depression, reports of self-harm and nonprescribed pain reliever use. On the other hand, higher levels of shame correlated with significantly worse reported health outcomes. Likewise, pride concepts composite scores had inverse and significant associations with anxiety and reports of self-harm.
As the authors point out, this study contributes a novel measure of shame and pride specific to the experiences of sexual minority adolescents.
However, this study has some methodological limitations that must be pointed out in the limitations section.
The authors used a stratified sampling. This is useful because this sampling method ensures specific groups are present in the sample. However, only 27% of the respondents were males. I do not understand why the authors did not select 50 % males and 50% females. On the other hand, the authors study sexual minority adolescents. Why 41% were bisexual, and only 16% were gay or lesbian, etc. Also, why 56% were white. This is not a probabilistic sample, is a purposeful sample. I understand that constructing a probabilistic sample would be impossible because this is a hidden population. However, the authors could have constructed a stratified sample (50 % males and 50% females; 33% white, 33% multiracial, 33% Hispanic; and a more balanced distribution of sexual minorities).
On the other hand, the authors used a respondent-driven sampling method. This sampling method is problematic because reproduces a homogeneous sample. Eligible participants identify individuals in their social network who think like them, have the same points of view, etc. I understand that when studying hidden populations respondent-driven sampling is the best way to access interviewees. However, the authors should have taken care of constructing a more balanced sample by gathering the demographic diversity of sexual minority adolescents.
Also, the sample size is quite small (n = 273). I think that this was a result of the budget the authors had.
Finally, the authors point out that the study was conducted in 18 months. However, they do not say when the study started (year-month) and when the fieldwork was concluded (year-month).
Author Response
The authors used a stratified sampling. This is useful because this sampling method ensures specific groups are present in the sample. However, only 27% of the respondents were males. I do not understand why the authors did not select 50 % males and 50% females. On the other hand, the authors study sexual minority adolescents. Why 41% were bisexual, and only 16% were gay or lesbian, etc. Also, why 56% were white. This is not a probabilistic sample, is a purposeful sample. I understand that constructing a probabilistic sample would be impossible because this is a hidden population. However, the authors could have constructed a stratified sample (50 % males and 50% females; 33% white, 33% multiracial, 33% Hispanic; and a more balanced distribution of sexual minorities).
Thank you for bringing this into conversation. The main study used stratified sampling based on geographic region, urbanicity, and sex. After completing the main survey, participants were asked if they wanted to complete an additional survey containing the shame and pride items for an additional incentive. Not all participants took advantage of this opportunity; so, our follow up is less balanced. We moved language from the measurement section to the participants and recruitment section (p. 5) and added language to the manuscript to make this clearer to readers:
“After completing the main study at this follow up, participants were asked if they wanted to complete an additional short survey containing the shame and pride items and general measures of shame and pride for testing concurrent criterion validity. Only a subset of participants from the main study completed the additional short survey”
On the other hand, the authors used a respondent-driven sampling method. This sampling method is problematic because reproduces a homogeneous sample. Eligible participants identify individuals in their social network who think like them, have the same points of view, etc. I understand that when studying hidden populations respondent-driven sampling is the best way to access interviewees. However, the authors should have taken care of constructing a more balanced sample by gathering the demographic diversity of sexual minority adolescents.
We appreciate the feedback. We included language in the limitations section (p. 12):
Our sampling methods are useful for recruiting marginalized people that are geo-graphically dispersed (Stern et al., 2020) but influences generalizability and interpre-tation of findings. We sampled only those with access to internet and who engage with social media. Furthermore, respondent-driven sampling can create homogeneity from the sample including like-minded individuals.
Also, the sample size is quite small (n = 273). I think that this was a result of the budget the authors had.
We appreciate your comment and acknowledge sample size constraints. We are close to the 300 cases cited by Tabachnick's rule of thumb, we are fair-good as suggested by Comrey and Lee, we exceed the recommended N=100 by Hair et al. we are close to the 10:1 ratio proposed, and we also exceed the recommended 5:1 ratio of Pallant (Williams, Onsman, & Brown, 2010). We can place sample size as a limitation if the reviewer would like but believe our sample size is adequate.
Williams, B., Onsman, A., & Brown, T. (2010). Exploratory factor analysis: A five-step guide for novices. Australasian journal of paramedicine, 8, 1-13.
Finally, the authors point out that the study was conducted in 18 months. However, they do not say when the study started (year-month) and when the fieldwork was concluded (year-month).
Data for this study were collected at 18 month follow up not over an 18 month span of time (p. 5).
Reviewer 2 Report
Comments and Suggestions for Authors
This paper introduces the Sexual Minority Identity Emotion Scale tailored for sexual minority adolescents, offering theoretical and practical contributions. Here are some suggestions for further refining the paper.
1. Line 32: Please elaborate on the impact of emotions on health outcomes. For example, how does the emotion of anger influence what kind of health outcomes (e.g., heart rate, etc.)?
2. Line 35: Please detail the physiological processes and behaviors that have been found to mediate the relationship between emotions and health outcomes.
3. Line 100: The items developed seem to diverge from the traditional emotion psychologists’ definition of emotion. The authors have created items reflecting beliefs, feelings, and goals associated with shame and pride through interviews. However, this broader definition strays from the conventional understanding of emotions as it incorporates elements like ‘beliefs’, which align more with cognition, and ‘goals’ or behavioral intentions, which are more akin to attitudes. It is advised that the authors engage more deeply with emotion literature (e.g., Lazarus, 1991), clarify the emotions of shame and pride, and differentiate them from cognitive and behavioral intentions. Furthermore, there is a need to align the broader emotion measurement used (including beliefs and goals) with the narrower definition presented in line 415 (‘Emotions are temporal feelings that give direction to thoughts and behaviors’) that emphasizes only the affective component.
Lazarus, R. S. (1991). Cognition and motivation in emotion. American psychologist, 46(4), 352.
4. Line 369: While Barrett (2017) is cited to support the argument that beliefs (cognition) and feelings (emotions) together contribute to emotional construction, a more in-depth, comprehensive theoretical explanation is needed. Additional references and detailed reasoning should be included to bolster the claim. This would strengthen the argument that cognition or beliefs can be considered a component of emotion, rather than just precursors.
Author Response
This paper introduces the Sexual Minority Identity Emotion Scale tailored for sexual minority adolescents, offering theoretical and practical contributions. Here are some suggestions for further refining the paper.
We appreciate your encouragement to think more deeply and better explicate the relationships between emotions and related mechanisms we discuss such as cognitions, behaviors and health. The feedback has strengthened our conceptualization and discussion of findings.
Line 32: Please elaborate on the impact of emotions on health outcomes. For example, how does the emotion of anger influence what kind of health outcomes (e.g., heart rate, etc.)?
Line 35: Please detail the physiological processes and behaviors that have been found to mediate the relationship between emotions and health outcomes.
Line 35 elaborates on Line 32. We have added the following language to provide examples of the way physiological processes and behaviors connect emotions to health (p., 1):
Fredrickson’s (2013) scholarship on positive emotions (e.g., joy, pride, hope) shows they relate to health through vagal tone, a physiological mechanism of the parasym-pathetic nervous system that regulates the body during changes. Conversely, negative emotions (e.g., anger, shame, and sadness), are related to health through a variety of physiological changes, such as increased heart rate and hormones related to the sympathetic nervous system (Cnsedine & Moskowitz, 2007). Emotions also elicit behaviors that have implications for health, including increased openness and social connections (positive emotions) or smoking, alcohol, other substance use or avoidance and isolation (negative emotions; Consedine & Moskowitz, 2007).
Line 100: The items developed seem to diverge from the traditional emotion psychologists’ definition of emotion. The authors have created items reflecting beliefs, feelings, and goals associated with shame and pride through interviews. However, this broader definition strays from the conventional understanding of emotions as it incorporates elements like ‘beliefs’, which align more with cognition, and ‘goals’ or behavioral intentions, which are more akin to attitudes. It is advised that the authors engage more deeply with emotion literature (e.g., Lazarus, 1991), clarify the emotions of shame and pride, and differentiate them from cognitive and behavioral intentions. Furthermore, there is a need to align the broader emotion measurement used (including beliefs and goals) with the narrower definition presented in line 415 (‘Emotions are temporal feelings that give direction to thoughts and behaviors’) that emphasizes only the affective component.
Lazarus, R. S. (1991). Cognition and motivation in emotion. American Psychologist, 46(4), 352.
Line 369: While Barrett (2017) is cited to support the argument that beliefs (cognition) and feelings (emotions) together contribute to emotional construction, a more in-depth, comprehensive theoretical explanation is needed. Additional references and detailed reasoning should be included to bolster the claim. This would strengthen the argument that cognition or beliefs can be considered a component of emotion, rather than just precursors.
We appreciate you raising this point for us to consider how we frame and define emotions in our study. Your feedback can be addressed by the methods and findings used to develop our measure. The qualitative roots of this measure used a Grounded Theory analysis that described the process of developing shame and pride related to one’s sexual minoritized identity (Goffnett, Routon & Flores, 2023). Findings show the importance of beliefs/cognitions and managing behaviors to the emotional experience of shame and pride. The original definitions separated beliefs, feelings, and behaviors in a way consistent with Lazarus and Barrett’s work. Where our measure deviates is by capturing the importance of this process to overall emotional construction of shame and pride. We have added the following language to the discussion (p., 11) to better clarify:
Findings form the qualitative study from which the SMIES emerged demonstrated the conditions relevant to feelings of shame and pride among sexual minority adoles-cents. The importance of beliefs and goals to the construction of these emotions was highlighted in the three dimension definition that guided the development of the measure’s items [8]. As such, the measure aimed to assess the construction of shame and pride through feelings and related beliefs and goals. Many theories of emotions high-light the importance of beliefs (cognitions) or goals (behaviors) to emotional generation and regulation [4,43]. By capturing the construction of shame and pride our measure brings beliefs and goals more central to the assessment of shame and pride deviating slightly from past approaches that consider these discrete processes
The line (415) “Emotions are temporal feelings that give direction to thoughts and behaviors,” was not intended to serve as a definition of emotions but to show that emotions influence thoughts a behaviors, and the interaction between these factors may help understand the role of emotions in health outcomes. Lazarus (1991) points to the cyclical relationship between cognitions and emotions, so we do not see this statement as being incongruent with our own work or past emotion theory.